# Intelligent Diagnosis of Rolling Element Bearing Based on Refined Composite Multiscale Reverse Dispersion Entropy and Random Forest

**DOI:** 10.3390/s22052046

**Published:** 2022-03-06

**Authors:** Aiqiang Liu, Zuye Yang, Hongkun Li, Chaoge Wang, Xuejun Liu

**Affiliations:** 1School of Mechanical Engineering, Dalian University of Technology, Dalian 116024, China; liuaq@mail.dlut.edu.cn (A.L.); liuxuejun@mail.dlut.edu.cn (X.L.); 2Microcyber Corporation, Shenyang 110179, China; yang.zuye@microcyber.cn; 3School of Logistics Engineering, Shanghai Maritime University, Shanghai 201306, China; cgwang@shmtu.edu.cn

**Keywords:** refined composite multiscale reverse dispersion entropy, random forest model, rolling bearing, intelligent diagnosis

## Abstract

Rolling bearings are the vital components of large electromechanical equipment, thus it is of great significance to develop intelligent fault diagnoses for them to improve equipment operation reliability. In this paper, a fault diagnosis method based on refined composite multiscale reverse dispersion entropy (RCMRDE) and random forest is developed. Firstly, rolling bearing vibration signals are adaptively decomposed by variational mode decomposition (VMD), and then the RCMRDE values of 25 scales are calculated for original signal and each decomposed component as the initial feature set. Secondly, based on the joint mutual information maximization (JMIM) algorithm, the top 15 sensitive features are selected as a new feature set and feed into random forest model to identify bearing health status. Finally, to verify the effectiveness and superiority of the presented method, actual data acquisition and analysis are performed on the bearing fault diagnosis experimental platform. These results indicate that the presented method can precisely diagnose bearing fault types and damage degree, and the average identification accuracy rate is 97.33%. Compared with the refine composite multiscale dispersion entropy (RCMDE) and multiscale dispersion entropy (MDE), the fault diagnosis accuracy is improved by 2.67% and 8.67%, respectively. Furthermore, compared with the RCMRDE method without VMD decomposition, the fault diagnosis accuracy is improved by 3.67%. Research results prove that a better feature extraction technique is proposed, which can effectively overcome the deficiency of existing entropy and significantly enhance the ability of fault identification.

## 1. Introduction

As an important component of large-scale electromechanical equipment, the rolling bearing health status is critical for the stable operation of equipment. Unfortunately, the rolling bearings are vulnerable components due to large load and unstable operation. Therefore, the intelligent diagnosis of rolling bearing is essential to improve the safety and stability of equipment, reduce maintenance costs, and avoid safety accidents [1,2,3].

The equipment will inevitably produce vibration during production operation, and the vibrations are the reflection of dynamic equipment characteristics. Therefore, the vibration signals contain a large amount of information characterizing the equipment’s health status, which has a clear physical meaning and can be easily identified [4]. At present, scholars have proposed various time-frequency analysis methods based on vibration signals, which can also be used to decompose signals. Common time-frequency analysis methods include the wavelet transform (WT) [5], resonance-based sparse signal decomposition (RSSD) [6], Hilbert vibration decomposition (HVD) [7], and empirical mode decomposition (EMD) [8], etc. However, the WT method needs to select wavelet base and decomposition layer beforehand. The EMD is prone to mode overlap and endpoint effects, and the RSSD method requires determining appropriate parameters in advance. The HVD is sensitive to additive noise and is not applicable for analyzing intermittent or non-oscillating signals, which makes them unable to obtain good application results [9,10,11]. To address these issues, Dragomiretskiy [12] presented the variational mode decomposition (VMD), which can achieve a perfect decomposition of non-linear, non-stationary signals. VMD has a solid theoretical background and analytical formulation has strong robustness and other advantages and has been shown to outperform EMD for intra-wave signal analysis [13,14]. Subsequently, VMD is widely used for rotating machinery fault diagnosis. In [15], VMD and energy entropy are combined for rolling bearing fault detection, which has a better effect than EMD and WT. In [16], the combination of VMD and Fourier synchro squeezed transform is used to extract bearing defect characteristics. In addition, VMD is also combined with deep learning methods such as convolution neural network [17], long short-term memory network [18], deep belief network [19] and plays a crucial role in fault diagnosis and remaining useful life prediction.

Due to the influence of complex operating conditions and environment, the collected vibration signals of the rolling bearing are usually non-stationary and non-linear [20]. Many non-linear dynamic methods, such as sample entropy [21], permutation entropy [22], fuzzy entropy [23], Rényi entropy [24], Wiener entropy [25], Instantantaneous Spectral Entropy [26], dispersion entropy [27], are proposed, which can reflect the non-linear properties of vibration signal and characterize equipment health status. Sample Entropy is slow in calculating long time series, poor in real-time performance, and prone to a sudden change in similarity measurement [28]. Although the calculation of permutation entropy is simple, the amplitude information of the time series is ignored [29]. Fuzzy entropy is an improvement of sample entropy, but there are still some problems, such as slow calculation speed [30]. Dispersion entropy has the advantage of being less affected by sudden change signals and solves the problems existing in permutation entropy with better stability [31]. However, due to the complexity of vibration signals, single-scale entropy cannot fully reflect fault information, thus multiscale signals can fully excavate fault information. Subsequently, the methods of time series multiscale [32], composite multiscale [33], refine composite multiscale [34] are gradually put forward by scholars. Among them, refined composite multiscale dispersion entropy (RCMDE) has better stability and feature extraction ability than multiscale dispersion entropy (MDE) [35]. In [36], a new feature for analyzing time series complexity, called reverse dispersion entropy, is put forward. The reverse dispersion entropy combines the advantages of permutation and dispersion entropy and has stronger noise robustness and stability. Inspired by refined composite multiscale, the refined composite multiscale reverse dispersion entropy (RCMRDE) is proposed in this article, which can mine rolling bearing fault information comprehensively.

To realize the automatic recognition of equipment health status, many intelligent diagnostic models, such as backpropagation neural network (BPNN), support vector machine (SVM), k-nearest neighbor (KNN), were extensively used. However, BPNN is easy to fall into local minimum [37], SVM is unable to deal with large-scale data [38], and KNN classification has high spatial complexity and poor robustness [39]. As a classification and regression tool, random forest (RF) can usually obtain excellent diagnostic accuracy in mechanical fault detection [40]. In [41], the refined composite hierarchical fuzzy entropy (RCHFE) and RF are utilized for planetary gearbox fault diagnosis. In [42], a fault diagnosis method based on core principal component analysis and RF is proposed, which is applied to a wind energy conversion system and achieves good results. Therefore, considering the excellent performance of RF, in order to accurately mine fault information from bearing monitoring data and achieve high precision fault pattern recognition, an intelligent fault diagnosis method integrating VMD decomposition, refine composite multiscale reverse dispersion entropy (RCMRDE), JMIM feature selection and RF is proposed. The stability, noise robustness, and signal discrimination ability of the proposed RCMRDE are verified by simulation signals. Furthermore, the effectiveness and superiority of the presented method are certificated by actual bearing vibration signals.

The main innovations of this research include the following two aspects:(1)RCMRDE is proposed for the first time, and its advantages in fault diagnosis are explored. Simulation and experimental results indicate that RCMRDE exhibits outstanding performance compared with several existing entropy.(2)There are few studies based on VMD and JMIM feature selection. JMIM feature selection can effectively calculate the resolution of each feature and select RCMRDE with high sensitivity to construct fault feature set. In this study, through JMIM feature selection, the original RCMRDE set is reduced by 91.4%, and the recognition accuracy is still 97.33%.

The remaining sections are organized as follows: Section 2 describes the implementation steps of RCMRDE and validates the superiority of RCMRDE by simulation case. Section 3 introduces the relevant theories and the proposed diagnostic framework. In Section 4, different rolling bearing health signals are collected, and then feature set construction based on VMD and RCMRDE and feature selection method based on JMIM algorithm are discussed. Next, some discussions and comparative analyses are carried out. Finally, the main contributions are summarized in Section 5.

## 2. Refined Composite Multiscale Reverse Dispersion Entropy

### 2.1. Reverse Dispersion Entropy

Reverse dispersion entropy [36] is used for detecting signal mutation, which combines the advantages of dispersion entropy and permutation entropy, and has better performance in mutation signal detection. Its calculation method is as follows:

Step1: Mapping time series to *c* classes.
(1)Mapping by the normal distribution function.

For the time series X={x(i),i=1,2,…,T} with T values, mapping X to Y={y(i),i=1,2,…,T} by Equation (1):(1)$$y_{i}=\frac{1}{\sigma \sqrt{2\pi }}\int_{-\infty }^{x_{i}}e^{\frac{-{\left(t-\mu \right)}^{2}}{2\sigma^{2}}}dt$$
where y(i)∈[0,1]. μ and σ denote expectation and variance, respectively.
(2)Using a linear algorithm to map each $y_{i}$ to integers in [1,c].

We map *Y* to Z={z(i),i=1,2,…,T} by using round $\left(c*y_{i}+0.5\right)$, where *c* is the category number.

Step 2: Using Equation (2) to calculate the embedding vectors, we reconstruct *Z* into *L*:(2)$$\left[\begin{array}{cccc} \{z(1), & z(1+\tau ), & \ldots & z\left(1+\left(m+1\right)d\right)\}\\ & \vdots & & \vdots \\ \{z(j) & z(j+\tau ), & \ldots & z\left(j+\left(m+1\right)d\right)\}\\ & \vdots & & \vdots \\ \{z(L) & z(L+\tau ), & \ldots & z\left(L+\left(m+1\right)d\right)\} \end{array}\right]$$
where *d* is the time delay and m is the embedding dimension, *L* is equal to T−(m−1)d.

Step 3: Mapping the dispersion pattern of each embedded vector.

There exist $c^{m}$ dispersion patterns, and each embedding vector can be mapped to a dispersion pattern π.

Step 4: Calculate the probability of each dispersion pattern.

The probability of *i*-th dispersion pattern can be written as:(3)$$P\left(\pi_{i}\right)=\frac{Number\left(\pi_{i}\right)}{N-\left(m-1\right)d}\left(1\leq i\leq c^{m}\right)$$
where $Number\left(\pi_{i}\right)$ is the number of mapping from each dispersion pattern to $\pi_{i}$, $P(\pi_{i})$ stands for the proportion of the number of *i*-th dispersion patterns to the number of embedding vectors.

Step 5: Calculating *RDE*.
(4)$$H_{RDE}\left(X,m,c,d\right)={\sum_{i=1}^{c^{m}}\left(P\left(\pi_{i}\right)-\frac{1}{c^{m}}\right)}^{2}=\sum_{i=1}^{c^{m}}P{\left(\pi_{i}\right)}^{2}-\frac{1}{c^{m}}$$

The normalized *RDE* is expressed as:(5)$$H_{RDE}=\frac{H_{RDE}\left(X,m,c,d\right)}{1-1/c^{m}}$$

### 2.2. Refined Composite Multiscale Reverse Dispersion Entropy

The refine processing of time series can mine statistical information more thoroughly than the coarse-grained process of multiscale algorithm, and the influence of the position of initial points on the calculation results can be effectively solved. RCMRDE includes the following steps:

For the time series $X=\left\{x_{1},x_{2},\ldots ,x_{T}\right\}$ of length *T*, the *k*-th coarse-grained series for a given scale factor *τ* can be given by:(6)$$x_{k,j}^{\tau }=\frac{1}{\tau }\sum_{i=\left(j-1\right)\tau +k}^{j\tau +k-1}x_{i},1\leq j\leq \left[T/\tau \right],1\leq k\leq \tau$$

For each scale factor *τ*, RCMRDE can be defined as:(7)$$RCMRDE\left(X,m,c,d,\tau \right)=-\sum_{i=1}^{c^{m}}\bar{P}{\left(\pi_{i}\right)}^{2}-\frac{1}{d^{m}}$$
where $\bar{P}\left(\pi_{i}\right)=\frac{1}{\tau }\sum_{1}^{\tau }P_{k}^{\left(\tau \right)}$, $P_{k}^{\left(\tau \right)}$ indicates the dispersion mode probability corresponding to the *k*-th coarsening sequence under scale τ.

### 2.3. Comparison between MDE, RCMDE, and RCMRDE Using Simulation Signals

In order to compare the entropy stability of RCMRDE, RCMDE, and MDE, we calculate the average entropy curves and error values (i.e., standard deviation) of RCMRDE, RCMDE, and MDE of 30 groups of white noise sequences under different data lengths and set *m* = 2, *d* = 1, *c* = 5, $\tau_{\max}$ = 25. The results are exhibited in Figure 1. It can be seen that the entropy curves of RCMRDE and RCMDE change more smoothly than MDE, which shows that the entropy estimation performance of RCMRDE and RCMDE is better than MDE. Meanwhile, compared with RCMDE and MDE, the standard deviation of RCMRDE is the smallest, which indicates that RCMRDE has outstanding stability.

To study the influences of data length on RCMRDE, three entropy values of white noise with different data lengths (N = 2048, 3072, 4096, 5120) were calculated, respectively. The parameters of the three entropy values were set as *m* = 2, *d* = 1, *c* = 5, $\tau_{\max}=25$. The results are shown in Figure 2. With the increase of τ, the curves of RCMRDE and RCMDE were smoother and had a smaller fluctuation range than MDE, which shows that RCMRDE and RCMDE have better stability than MDE. Meanwhile, the proposed RCMRDE can exhibit good feature extraction ability under less data length.

Subsequently, in order to evaluate the calculation complexity of the three entropy values, we used WGN noise with different data lengths (*N* = 2048, 3072, 4096, 5120) to calculate the computing times they spent. The results are listed in Table 1. The parameters of the three kinds of entropy were set as *m* = 2, *d* = 1, *c* = 5, $\tau_{\max}=25$. Table 1 shows that the calculation time of RCMRDE was slightly longer than MDE but largely lower than RCMDE. With the increase of data length, the calculation time of three entropy values increased significantly. Therefore, selecting less data length can improve the computational efficiency when extracting features.

Furthermore, in order to research the influences of category number c on RCMRDE, three entropy values of WGN (MDE, RCMDE, and RCMRDE) were calculated under different values. The results are displayed in Figure 3. The parameters of the three entropy values were set as *m* = 2, *d* = 1, $\tau_{\max}=25$, *n* = 2048, and c increased from 3 to 7. It can be found that compering with the MDE curves, the RCMDE curves and the RCMRDE curves fluctuated slightly (except c=3), which indicates that RCMDE and RCMRDE can provide stable and reliable entropy values under different *c*.

In addition, we can see from Figure 3c that with the increase of *c* value, the dispersion mode will increase, thus leading to the decrease of RCMRDE curves. Meanwhile, in practical application, if the *c* value was too small, the signal characteristic information cannot be fully extracted. Therefore, considering the reliability and calculation efficiency, the *c* value should be set as 5 or 6.

When local damage occurs to rolling bearings, the measured vibration signal contains much noise. Therefore, we used Equation (8) to study the anti-noise performance of RCMRDE.
(8)$$x(t)=[1+0.5\cos(8\pi t)]\cos[200\pi t+2\cos(10\pi t)]+0.8\sin(\pi t)\sin(30\pi t)+n(t)t\in (0,1)$$
where n(t) represents the Gaussian white noise with a SNR of −5, 0, and 5 dB, respectively. In addition, *m*, *c*, *d* and $\tau_{\max}$ of MDE, RCMDE, and RCMRDE are set to 3, 5, 1, and 25, respectively. The signal length was 2000 points. Figure 4 displays the calculation results of three entropy values. With the increase of τ, the RCMRDE value was more concentrated and the fluctuation was smaller, which proves that RCMRDE has excellent noise robustness.

Herein, the bearing outer race fault model with four different fault degrees was established to validate the superiority of RCMRDE in signal discrimination ability. The mathematical expression of the fault model is as follows:(9)x(t)=e(t)+r(t)+p(t)+n(t)
where e(t) denotes a repetitive pulse caused by a local defect:(10)$$e\left(t\right)=\sum_{i=1}^{M_{1}}A\left(t\right)e^{-\zeta_{\alpha }\left(t-iT_{\alpha }-\delta_{i}\right)}\cos\left[2\pi f_{\alpha }\left(t-iT_{\alpha }-\delta_{i}\right)+\varphi_{\alpha }\right]$$
where $M_{1}$ represents the number of fault pulses and A(t) stands for the amplitude of fault impulses. $T_{\alpha }$ is the time interval between two adjacent fault impulse. $\delta_{i}$ is a random time lag between two repetitive impulses. $f_{\alpha }$, $\zeta_{\alpha }$ and $\varphi_{\alpha }$ are the resonant frequency, damping coefficient, and phase excited by fault impacts, respectively.

In Equation (9), r(t) stands for random impulses generated by external excitation:(11)$$r\left(t\right)=\sum_{s=1}^{M_{2}}B_{s}e^{-\zeta_{b}\left(t-T_{s}\right)}\cos\left[2\pi f_{b}\left(t-T_{s}\right)+\varphi_{b}\right]$$
where $M_{2}$, $B_{s}$ and $T_{s}$ represents the number, amplitude, and occurrence time of random impulses, respectively. The meanings of $f_{b}$, $\zeta_{b}$ and $\varphi_{b}$ the same as those of the corresponding symbols in Equation (10).

In Equation (9), p(t) is the pure periodic component generated by axis rotation:(12)$$p\left(t\right)=\sum_{k=1}^{M_{3}}C_{k}\sin\left(2\pi f_{k}t+\theta_{k}\right)$$
where $M_{3}$ is the number of periodic harmonics. $C_{k}$, $f_{k}$ and $\theta_{k}$ are the amplitude, frequency, and phase of components, respectively. n(t) in Equation (9) is the added noise. The model parameters are given in Table 2.

In the simulation experiment, the sampling frequency was 12,800 Hz, and the signal length was 8192 points. N (2.5, 1) and U (1, 8192) denoted a normal distribution and a uniform distribution, respectively. Adjust the size of A(t) to 0.5, 0.6, 0.7, and 0.8, respectively. Therefore, four simulation signals that represented by $X_{1}$, $X_{2}$, $X_{3}$, $X_{4}$ were generated, corresponding to the damage degree of the outer ring from slight to serious. The waveforms of four simulation signals are given in Figure 5a. However, the fault types and degrees cannot be distinguished by the time domain waveform. Next, we calculated the MDE, RCMDE, and RCMRDE of simulation signals with four different fault degrees, as displayed in Figure 5.

It can be seen from Figure 5 that the four curves of MDE intersect with each other, while RCMDE and RCMRDE show good independence, which indicates that RCMDE and RCMRDE have outstanding fault discrimination ability. Furthermore, RCMRDE shows better distinguishing performance than RCMDE in determining the degree of bearing fault. Thus, the RCMRDE can more accurately characterize the bearing health status.

## 3. The Proposed Fault Diagnosis Method

### 3.1. Variational Mode Decomposition

VMD is an adaptive non-stationary signal decomposition technology, which can decompose the given signal into an ensemble of intrinsic mode functions (IMFs), and each IMF has limited bandwidth in its spectrum [43]. VMD determines the correlation frequency band and decomposes the original signal into k modal components. The constrained variational problems are given by Equation (13):(13)$$\begin{array}{l} \underset{\left\{u_{k}\right\},\left\{\omega_{k}\right\}}{\min}\left\{\sum_{k}{\left\|\partial_{t}\left[\left(\delta \left(t\right)+\frac{j}{\pi t}\right)*u_{k}\left(t\right)\right]e^{-j\omega_{k}t}\right\|}_{2}^{2}\right\}\\ s.t.\sum_{k}u_{k}=f\left(t\right) \end{array}$$
where $u_{k}$ is the *k*-th IMF of the signal. $\omega_{k}$ is the center frequency. f(t) is the input signal and δ(t) is the Dirac function. $\partial_{t}$ is the partial derivative of a function to *t*. ∗ is the convolution symbol. Considering a quadratic penalty term and Lagrange multipliers λ, Equation (13) can be written as follows:(14)$$\begin{array}{l} L\left(\left\{u_{k}\right\},\left\{\omega_{k}\right\},\lambda \right)=\alpha \sum_{k}{\left\|\partial_{t}\left[\left(\delta \left(t\right)+\frac{j}{\pi t}\right)*u_{k}\left(t\right)\right]e^{-j\omega_{k}t}\right\|}_{2}^{2}\\ +{\left\|f\left(t\right)-\underset{k}{\sum }u_{k}\left(t\right)\right\|}_{2}^{2}+\langle \lambda \left(t\right)f\left(t\right)-\sum_{k}u_{k}\left(t\right)\rangle \end{array}$$
where α is the balance parameter of the data fidelity constraint. λ is a common method of strictly enforcing constraints. ${\left\|•\right\|}_{2}^{2}$ represent the squared $L^{2}$ norm. VMD employs the alternating direction method of multipliers to solve Equation (14). The solution of the second-order optimization problem can be obtained. Each estimated IMF can be expressed as:(15)$${\hat{u}}_{k}^{n+1}\left(\omega \right)=\frac{\hat{f}\left(\omega \right)-\sum_{i\neq k}{\hat{u}}_{k}^{n}\left(\omega \right)+\frac{\hat{\lambda }\left(\omega \right)}{2}}{1+2\alpha {\left(\omega -\omega_{k}\right)}^{2}}$$
where $\hat{f}(\omega )$, ${\hat{u}}_{i}(\omega )$, $\hat{\lambda }(\omega )$ stand for the Fourier transforms of f(ω), $u_{i}(\omega )$ and λ(ω), respectively. n represents the iterations number. As above, optimization is performed in the Fourier transform domain to find the optimal central frequency $\omega_{k}$, which can be obtained by Equation (16). The acquired new center frequency is given as:(16)$$\omega_{k}^{n+1}=\frac{\int_{0}^{\infty }\omega^{2}{\left|{\hat{u}}_{k}\left(\omega \right)\right|}^{2}d\omega }{\int_{0}^{\infty }{\left|{\hat{u}}_{k}\left(\omega \right)\right|}^{2}d\omega }$$

### 3.2. Feature Selection Based on JMIM

Feature selection is widely used in many fields, such as data mining and machine learning. Feature selection based on information theory is a popular method because it has a tremendous advantage in computational efficiency. However, the disadvantage of this method is lacking information about the interaction between features and classifiers, as well as the selection of redundancy features. Therefore, it is crucial to select a reasonable feature selection method to reduce dimension and improve classification accuracy.

The joint mutual information maximization (JMIM) is an effective feature selection algorithm, which can extract features and create a feature subset efficiently based on joint mutual information [44]. Compared with many other feature selection methods such as joint mutual information (JMI), maximum relevancy minimum redundancy (mRMR), etc. JMIM feature selection method can achieve the best trade-off in accuracy and stability [45]. Therefore, this paper employs JMIM as the sensitive feature selection criterion to reduce features redundancy and improve recognition accuracy.

### 3.3. Random Forest

Random Forest (RF) is a machine learning algorithm that consists of many independent decision trees [46]. RF uses multiple CART (Classification and Regression Tree) as meta-classifier and applies bagging algorithms to produce different training sample sets. Meanwhile, it randomly selects features to split the internal nodes when constructing a single tree. Therefore, RF can tolerate noise better and have better classification performance. As a multi-functional machine learning algorithm in practical application, RF is not only used for regression and classification but also for processing missing values, outliers, and other data exploration. The RF algorithm includes the following steps [47]:

Step 1: set a sample set of N for the samples number, M for the variables number;

Step 2: each node will randomly select *m* (*m < M*), a specific variable, which is then used to determine the optimal splitting point. The value of *M* remains constant during the generation of the decision tree;

Step 3: sample N times from the sample set (*N* samples) to form a set of training sets;

Step 4: for each node, m variables based on this point are randomly selected, and their optimal splitting points are calculated;

Step 5: each decision tree will grow as much as possible without pruning and will forecast new data by adding all the decision trees together.

### 3.4. Proposed Fault Diagnostic Framework

In this paper, an intelligent diagnosis method integrated VMD, RCMRDE, and RF model is shown in Figure 6, and specific implementation steps include:(1)VMD is applied to decompose the original signal into several modal components. The modal number is based on the decomposition criterion that the frequency center frequency of each component is not overlapping.(2)Based on the original signal and VMD decomposed components, RCMRDE at 25 scales is calculated as the initial feature set. The high-dimensional features contain redundant information. Subsequently, JMIM is employed to select sensitive features, thereby removing redundant information and reducing the dimension of feature set data.(3)Input sensitive features selected in steps (2) into the RF model to identify bearing health status. The presented method performance is checked by the rolling bearing vibration signals under different conditions.

## 4. Experiments and Data Analysis

### 4.1. Experimental Setup and Data Acquisition

To verify the effectiveness and feasibility of the presented method, actual data collection was performed using the bearing fault diagnosis test bench QPZZ-II, which is displayed in Figure 7a. The test bench can simulate multiple bearing failure types. In order to simulate the bearing fault damage in the real situation, 2 different types of bearings were processed by wire cutting. The fault types were inner race 0.2 mm, 0.4 mm, and 0.6 mm wear, outer race 0.2 mm, 0.4 mm, and 0.6 mm crack, and rolling element 0.2 mm, 0.4 mm, and 0.6 mm wear. The bearings of the 3 fault types are shown in Figure 7b. The sensor used 3035B accelerometer produced by the DYTRAN company in the United States, which was installed in the radial direction of the rolling bearing. The sampling frequency was 12,800 Hz, and the rotating speed was 1200 r/min. The bearing models used were N205EM/PS with a detachable outer ring and NU205EM/PS with a detachable inner ring. A total of 50 non-overlapping samples were selected from the vibration signals of each fault type, and the length of each sample was 2048 points, of which 35 groups were used for training and 15 groups were used for testing. The dimensional parameters of the bearing are shown in Table 3. Table 4 gives the description of experimental data. Bearing vibration signals under 10 different health conditions are illustrated in Figure 8. One cannot judge the fault types and damage degrees only by observing the time domain waveforms. Therefore, it was necessary to select reasonable feature extraction and recognition methods to realize the automatic and accurate discrimination of bearing health states.

### 4.2. Feature Extraction by VMD-Based RCMRDE

In this section, the VMD was applied to decompose the original signal into multiple IMF components. The principle of setting the number of VMD decomposition was that the center frequency of each component spectrum will not overlap after decomposition. When the decomposition level was set to 6, all 10 types of signals were decomposed sufficiently without overlapping of the center frequency of each component. Therefore, the decomposition level was set to 6. Here, taking the normal bearing signal as an example, its frequency spectrum and corresponding decomposition components are shown in Figure 9. It can be found that the original signal was decomposed into 6 IMF components, which achieved complete decomposition and no frequency aliasing. Subsequently, the parameters of RCMRDE were set as *m* = 2, *d* = 1, *c* = 5, $\tau_{\max}=25$. The VMD decomposition results of each fault type signal are given in Figure 10.

According to Steps 2 in Section 3.4, there were 25 RCMRDE values obtained from the original signal, and 150 RCMRDE values were obtained from the VMD decomposed components. Therefore, a total of 175 eigenvalues were obtained for each type of fault signal. The RCMRDE values of the original signal and the first IMF component are displayed in Figure 11. We can observe that the RCMRDE values of different fault types were well separable, which indicates the proposed method can mine the distinguishing characteristics between different types and degrees of bearing fault.

### 4.3. Diagnosis Results and Analysis

Considering that high-dimensional features data contain redundant information, the JMIM method is utilized to select the sensitive RCMRDE values with large contributions and excellent classification performance. Finally, the top 15 features selected are used for establishing the new feature set and then feed them into the RF model. Figure 12 depicts the multi-class confusion matrix of the presented method. It can be observed that the classification accuracy of ORF02, ORF06, REF02, and REF06 was 93.33%, and the other fault types were 100%, and the average classification accuracy was 97.3%, which indicates that the proposed method can accurately and effectively identify bearing health status and obtain satisfactory diagnosis results.

Furthermore, in order to illustrate the superiority of the presented feature extraction algorithm, the RCMRDE features were compared with RCMDE and MDE, and the entropy values of the top 15 features were selected, respectively, and then input into the RF model. The diagnostic accuracy is given in Figure 13. It can be known that the diagnostic accuracy of RCMRDE was the best, up to 97.33%, followed by RCMDE, up to 94.67%. Meanwhile, with the increase of features number, the diagnostic accuracy of RCMRDE shows the fastest growth rate. The above results indicate that RCMRDE can more thoroughly detect the dynamic mutation of the bearing fault signal.

Subsequently, the JMIM is compared with Fisher [48] and LS [49]. Figure 14 illustrates the diagnostic accuracy of the 3 methods. We can see from Figure 14 that the presented method achieves the highest diagnostic accuracy of 97.33% (12 sensitive features are selected), the Fisher method obtained the diagnostic accuracy of 94.67% (14 sensitive features are selected), and the LS method gets the diagnostic accuracy of 90.67% (14 sensitive features are selected). Some conclusions can be drawn through comparison: (1) the sensitive feature selection criteria can reduce the dimension of feature data and improve computational efficiency; (2) JMIM is better than Fisher and LS in terms of sensitive feature selection; (3) the effective sensitive features can significantly improve diagnostic accuracy.

### 4.4. Comparison with Other Methods

To further evaluate the presented method, some machine learning algorithms, such as KNN, BPNN, and SVM, were selected for comparison. In total, the proposed method was compared with the other 23 methods, and the results are shown in Figure 15. In the figure, the MDE_ Original stands for extracting MDE features from the original signal, where $\tau_{\max}$ is set to 25 and the top 15 sensitive features are selected as the final feature set. It should be noted the feature set construction process in RCMDE_ Original and RCMRDE_Original are the same as those in MDE_Original. In addition, the MDE_VMD stands for extracting MDEs from the original signal and the VMD decomposition components, where the $\tau_{\max}$ of the original signal and each IMF were set to 25, the decomposition number of VMD was set to 6. Then, the top 15 features were selected from 175 features data to build the sensitive feature set. Similarly, the feature set construction process in RCMDE_VMD and RCRMDE_VMD were the same as those in MDE_VMD.

It can be found from Figure 15 that when using the same machine learning model (except BPNN), the accuracy of RCMRDE_VMD was higher than or equal to MDE_VMD and RCMDE_VMD. This proves that RCMRDE can more clearly characterize the bearings health status. In addition, for the 4 machine learning models, the diagnostic accuracy of MDE_VMD was better than MDE_Original, RCMDE_VMD was better than RCMDE_Original, and RCMRDE_VMD was better than RCMRDE_Original. This indicates that the sensitive feature information can be more fully excavated after the original signal is decomposed by VMD. It is worth noting that the diagnostic accuracy of the presented method was highest, with a recognition rate of 97.33%. As a result, the presented method has a superior performance in the intelligent diagnosis of bearing fault type and damage degree.

## 5. Conclusions

In this research, a novel intelligent diagnosis of rolling bearings combining RCMRDE and RF model is proposed. First, aiming at the disadvantage of traditional multiscale entropy cannot accurately extract the useful feature from non-stationary fault signal, RCMRDE is created to characterize different fault types and damage degrees of rolling bearings. The simulated experiment shows that RCMRDE performs better in mutation signal detection, noise robustness test, stability, and signal discrimination. Secondly, the JMIM method is applied to select sensitive features and build feature datasets, thus reducing redundant information and improving computational efficiency. Then, the sensitive feature set is fed into the RF model to realize the automatic and accurate discrimination of bearing health status. The effectiveness of the proposed method is fully demonstrated by the simulation signals and actual bearing diagnosis experiment. Furthermore, the proposed method shows superior performance compared with other machine learning methods and feature extraction methods. The comparisons show that the proposed method achieves outstanding diagnostic results, and its recognition accuracy is 97.33%. However, the decomposition number of VMD still needs further optimization. Future work will consider choosing a good optimization algorithm to realize the best mode decomposition of the rolling bearing fault signal.

## Figures and Tables

**Figure 1 sensors-22-02046-f001:**
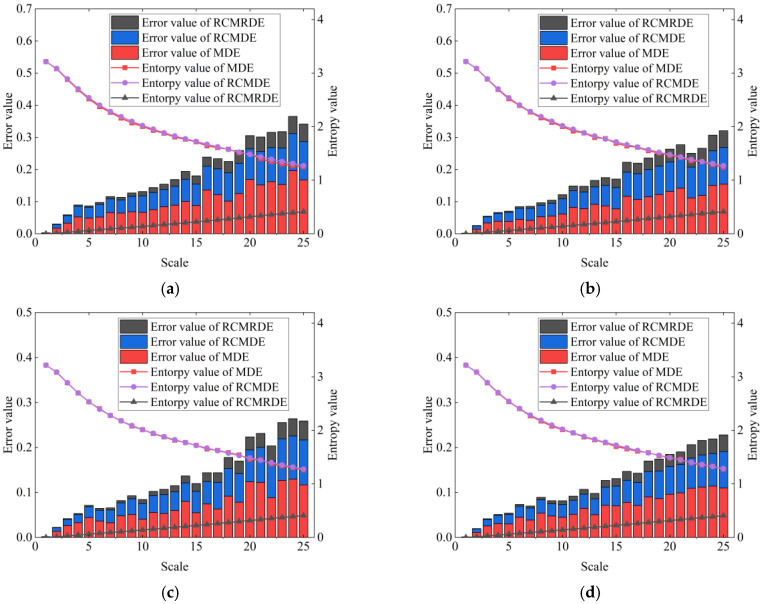
The mean curve and error value of MDE, RCMDE and RCMRDE for white noise with different N: (**a**) N = 2048; (**b**) N = 3072; (**c**) N = 4096; (**d**) N = 5120.

**Figure 2 sensors-22-02046-f002:**
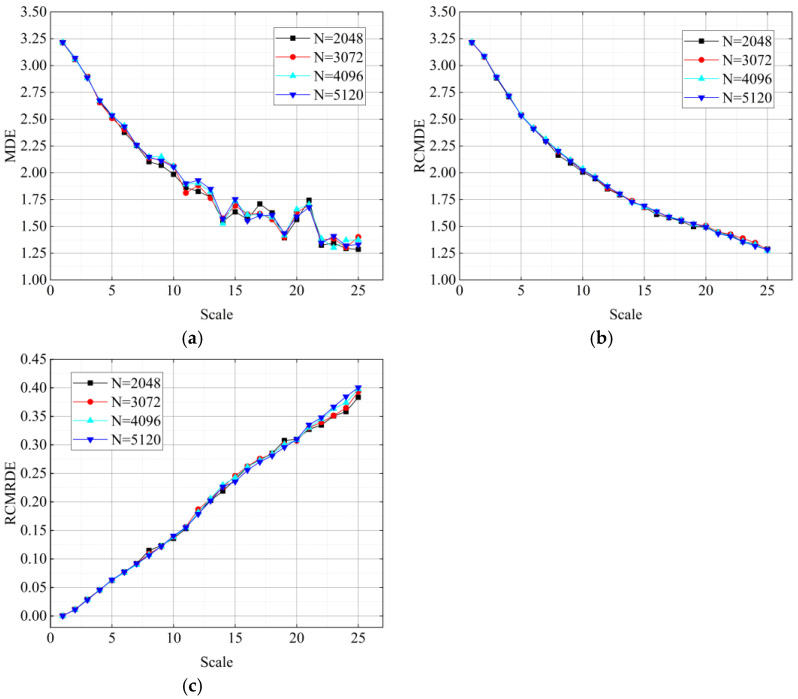
The MDE, RCMDE, and RCMRDE of WGN noise under different data lengths: (**a**) MDE; (**b**) RCMDE; (**c**) RCMRDE.

**Figure 3 sensors-22-02046-f003:**
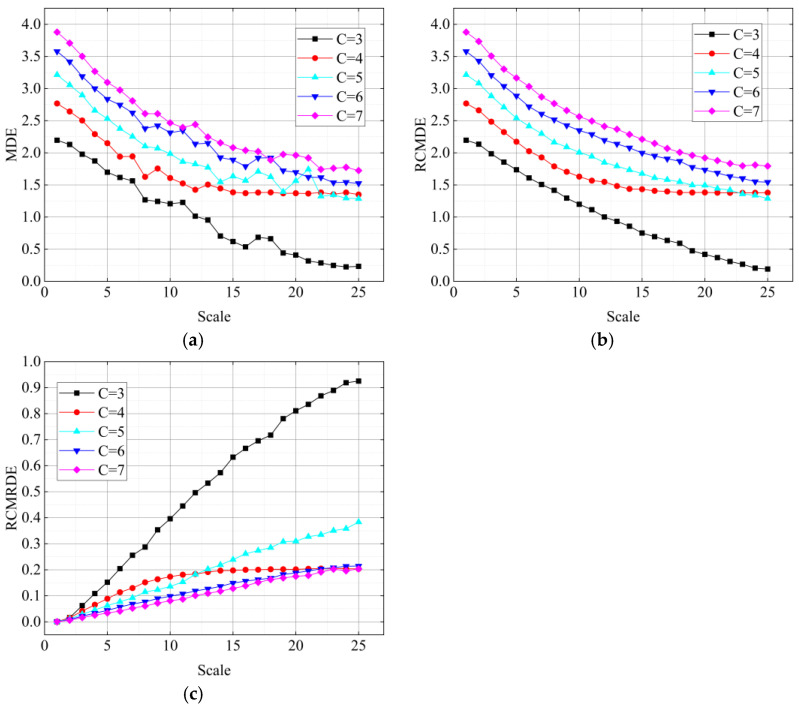
MDE, RCMDE, RCMRDE of WGN noise under different numbers of c (**a**) MDE of WGN; (**b**) RCMDE of WGN; (**c**) RCMDE of WGN.

**Figure 4 sensors-22-02046-f004:**
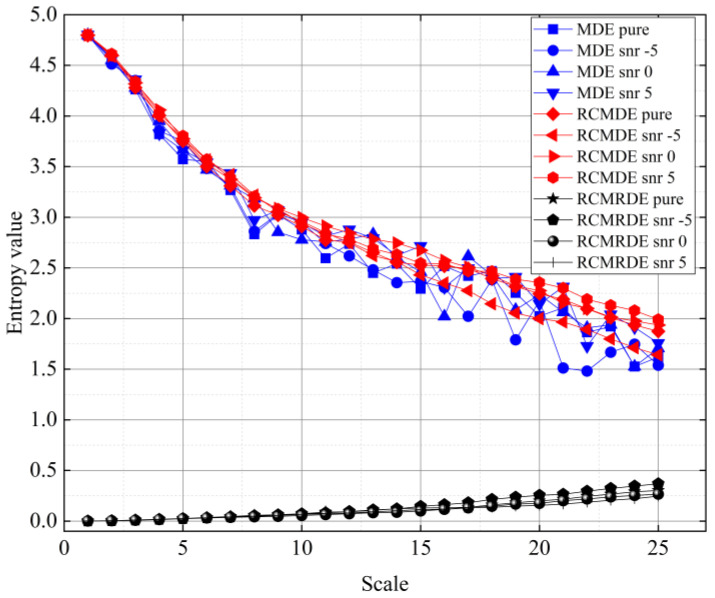
The MDE, RCMDE, and RCMRDE of synthetic signal with different SNRs.

**Figure 5 sensors-22-02046-f005:**
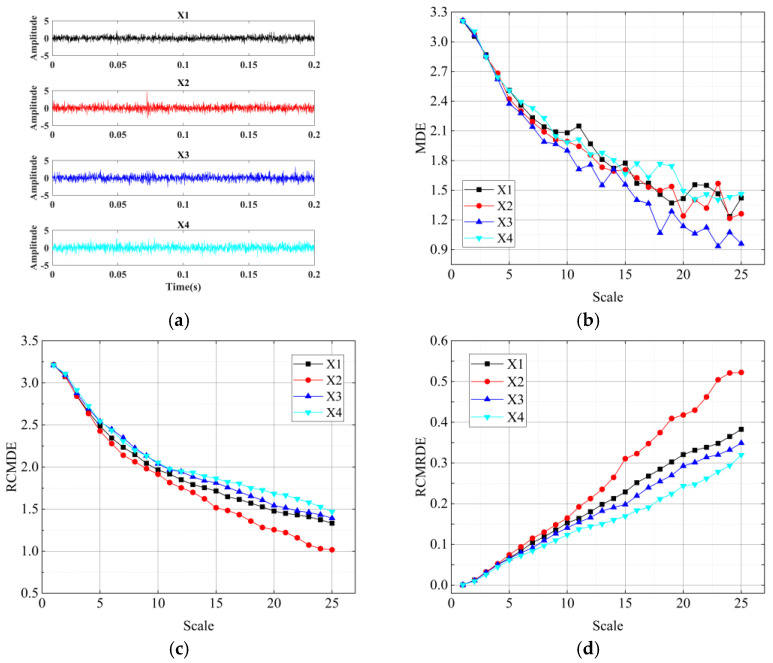
Simulation signals and corresponding three entropy values: (**a**) four bearing outer fault signals with different damage degrees; (**b**) MDE value curves; (**c**) RCMDE value curves; (**d**) RCMRDE value curves.

**Figure 6 sensors-22-02046-f006:**
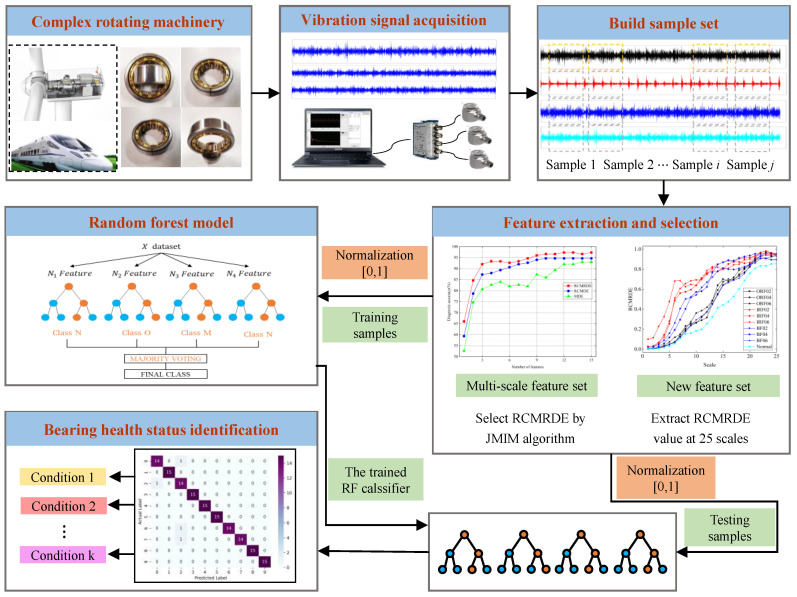
The flowchart of the proposed method.

**Figure 7 sensors-22-02046-f007:**
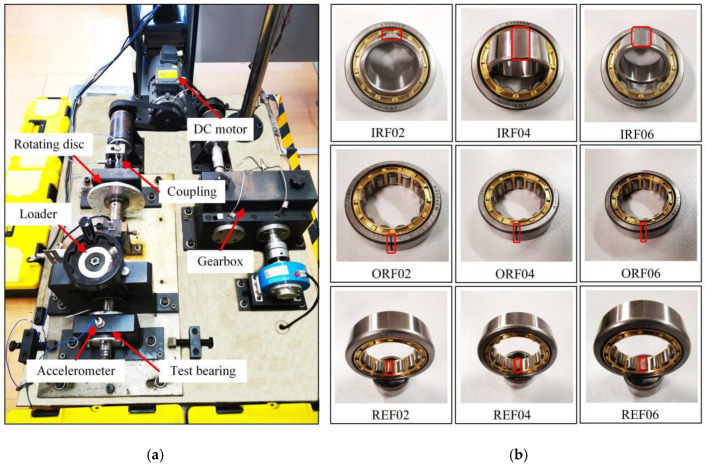
Rolling bearing fault diagnosis test bench and faulty parts: (**a**) test bench; (**b**) different types of faulty parts.

**Figure 8 sensors-22-02046-f008:**
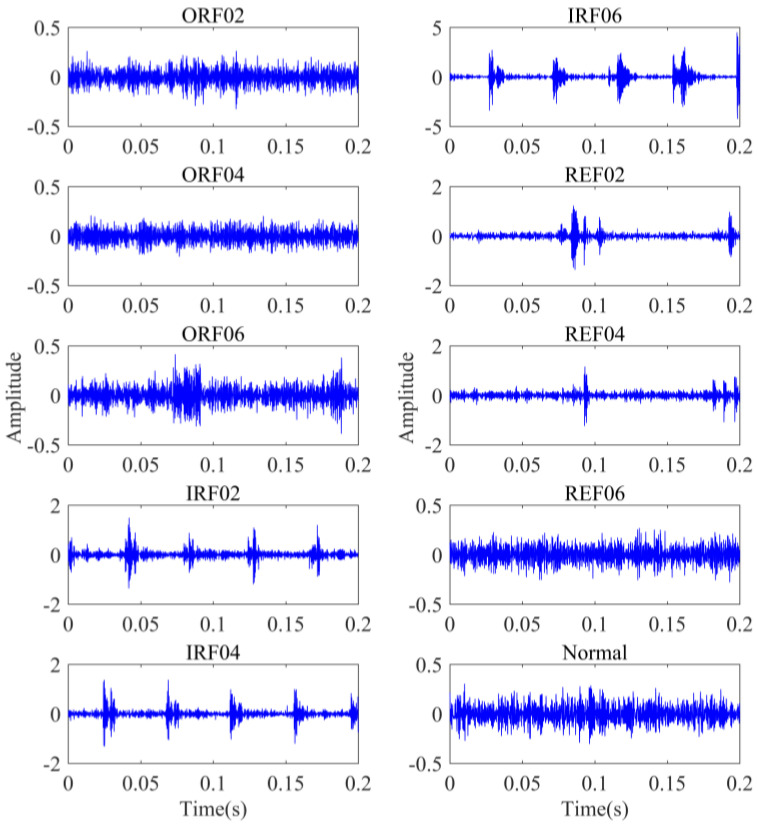
Bearing vibration signals under 10 different health conditions.

**Figure 9 sensors-22-02046-f009:**
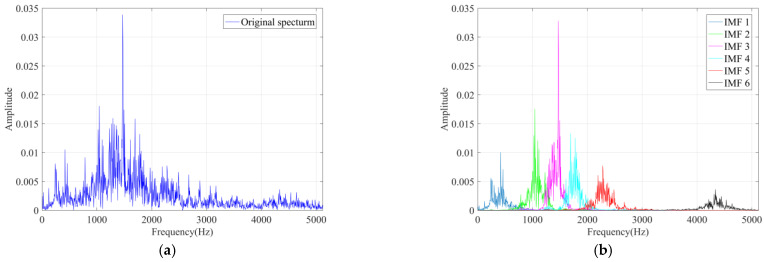
The frequency spectrum of normal bearing signal: (**a**) spectrum of original signal; (**b**) spectrum of each component.

**Figure 10 sensors-22-02046-f010:**
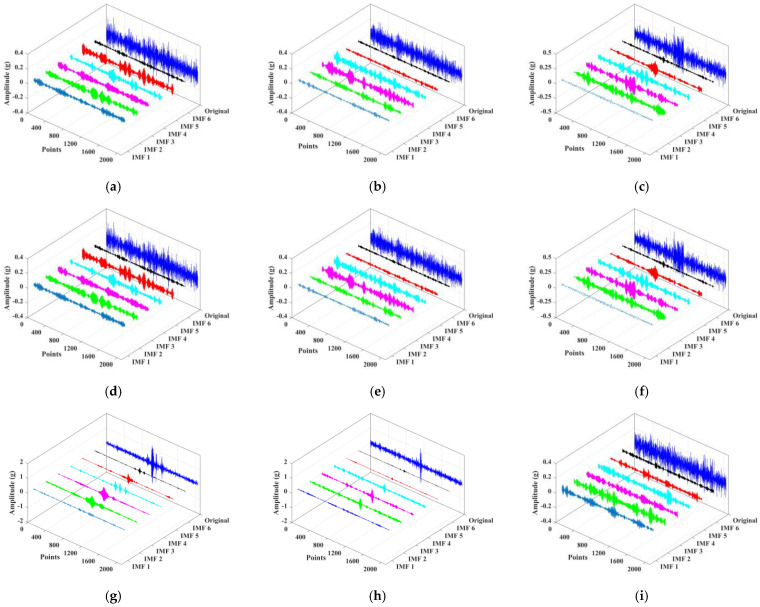
The VMD decomposition results of each fault type signal: (**a**) outer race 0.2 mm crack; (**b**) outer race 0.4 mm crack; (**c**) outer race 0.4 mm crack; (**d**) inner race 0.2 mm wear; (**e**) inner race 0.4 mm wear; (**f**) inner race 0.6 mm wear; (**g**) rolling element 0.2 mm wear; (**h**) rolling element 0.4 mm wear; (**i**) rolling element 0.6 mm wear.

**Figure 11 sensors-22-02046-f011:**
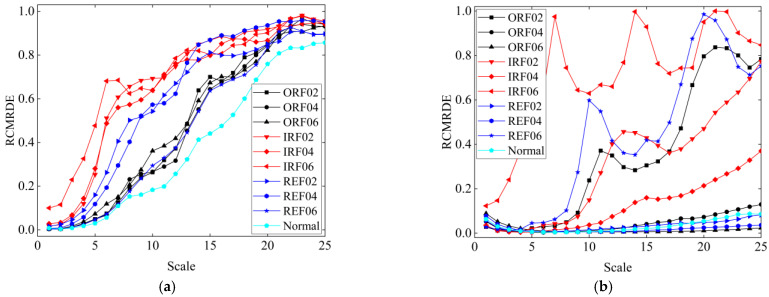
The RCMRDE values obtained under scale 25: (**a**) RCMRDE values of original signal; (**b**) RCMRDE values of the first IMF components.

**Figure 12 sensors-22-02046-f012:**
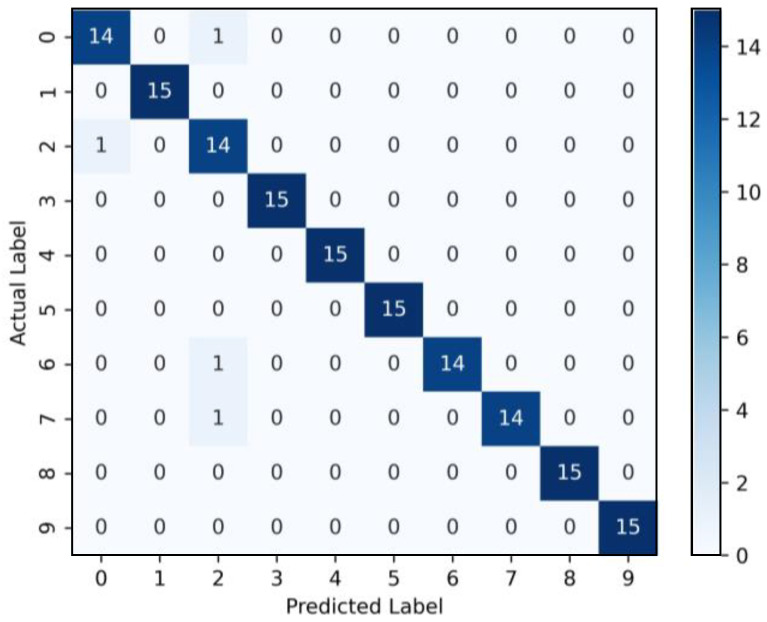
Multi-class confusion matrix of the proposed method.

**Figure 13 sensors-22-02046-f013:**
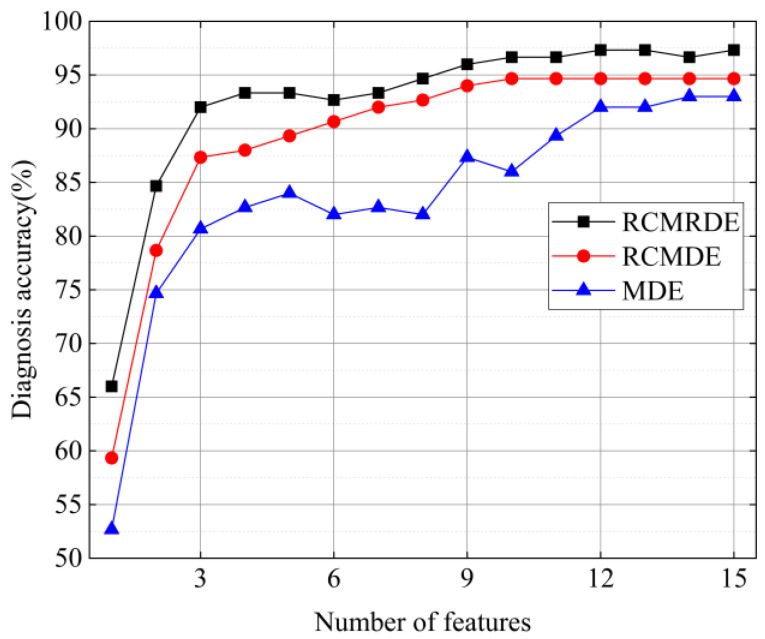
The diagnosis accuracy of three methods.

**Figure 14 sensors-22-02046-f014:**
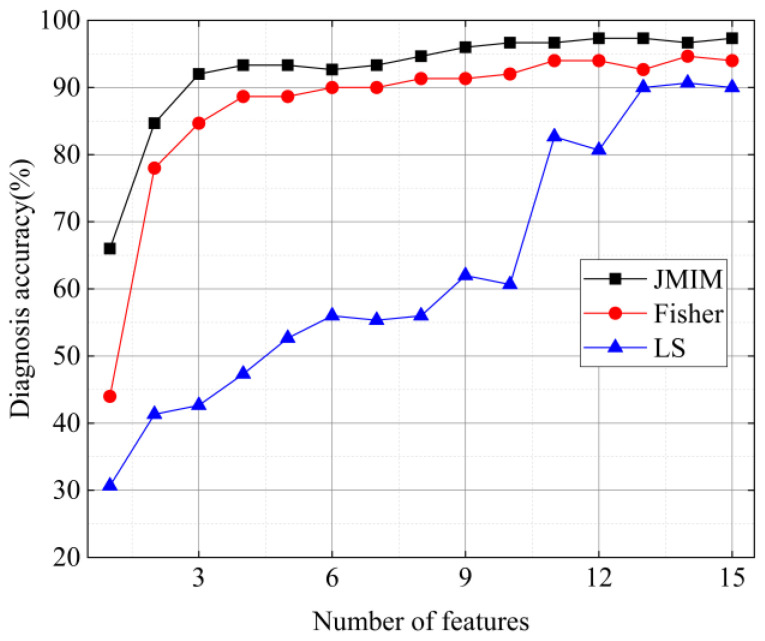
The diagnosis accuracy of JMIM, Fisher and LS feature selection methods.

**Figure 15 sensors-22-02046-f015:**
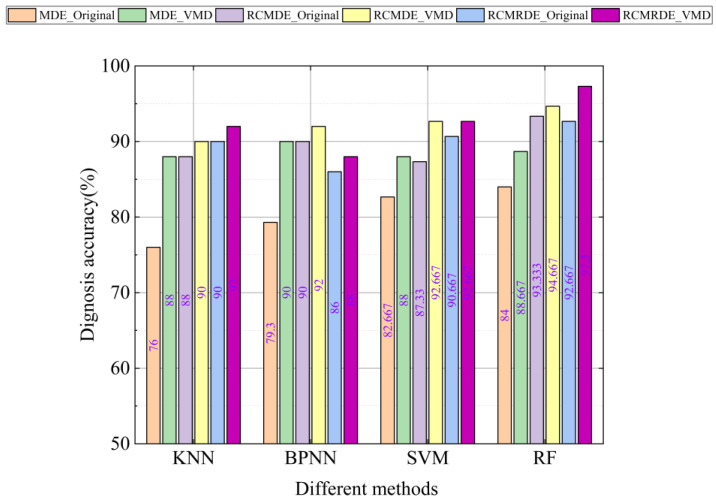
Comparison of diagnostic accuracy of different methods.

**Table 1 sensors-22-02046-t001:** The comparison of computation time of three entropy values under different data lengths.

Methods	Data Length			
	2048	3072	4096	5120
MDE	0.1059 s	0.1148 s	0.1235 s	0.1335 s
RCMDE	0.2061 s	0.2867 s	0.3704 s	0.4490 s
RCMRDE	0.1997 s	0.2818 s	0.3637 s	0.4431 s

**Table 2 sensors-22-02046-t002:** Parameter values of bearing outer race fault simulation model.

Parameter	Value	Parameter	Value	Parameter	Value
$M_{1}$	73	$M_{2}$	3	$M_{3}$	2
$C_{2}\left(m/s^{2}\right)$	0.025	$B_{s}$	N (2, 5, 1)	$C_{2}\left(m/s^{2}\right)$	0.025
$A\left(t\right)\left(m/s^{2}\right)$	0.5, 0.6, 0.7, 0.8	$T_{s}\left(s\right)$	$U\;(1,\;8192)/f_{s}$	$f_{1}\left(Hz\right)$	10
$f_{\alpha }\left(Hz\right)$	2600	$f_{b}\left(Hz\right)$	1700	$f_{2}\left(Hz\right)$	20
$\varphi_{\alpha }\left(rad\right)$	0	$\varphi_{b}\left(rad\right)$	0	$\theta_{1}\left(rad\right)$	π/6
$\zeta_{\alpha }$	1000	$\zeta_{b}$	800	$\theta_{2}\left(rad\right)$	−π/3

**Table 3 sensors-22-02046-t003:** The information of bearing dimension.

Type	Inner Diameter (mm)	Outer Diameter (mm)	Rolling Element Diameter (mm)	Pitch Circle Diameter (mm)	Number of Rolling Element	Characteristic
N205EM∕PS	25	52	7.5	39	13	Detachable outer ring
NU205EM∕PS	25	52	7.5	39	13	Detachable inner ring

**Table 4 sensors-22-02046-t004:** The description of experimental data.

Bearing State	Fault Size (mm)	Abbreviation	Label	Training Data	Test Data
Outer race fault	0.2	ORF02	0	35	15
Outer race fault	0.4	ORF04	1	35	15
Outer race fault	0.6	ORF06	2	35	15
Inner race fault	0.2	IRF02	3	35	15
Inner race fault	0.4	IRF04	4	35	15
Inner race fault	0.6	IRF06	5	35	15
Rolling element fault	0.2	REF02	6	35	15
Rolling element fault	0.4	REF04	7	35	15
Rolling element fault	0.6	REF06	8	35	15
Normal	-	Normal	9	35	15

## Data Availability

The data presented in this study are available on request from the corresponding author. The data are not publicly available due to privacy.

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
