# Peer review of "Intelligent Diagnosis of Rolling Element Bearing Based on Refined Composite Multiscale Reverse Dispersion Entropy and Random Forest"

_sensors, 2022, doi:10.3390/s22052046_

Round 1
Reviewer 1 Report
In this research, a novel intelligent diagnosis of rolling bearings combining RCMRDE and RF model is proposed.
The simulated experiment shows that RCMRDE performs better in mutation signal detection, noise robustness test, stability and signal discrimination.
The JMIM method is applied to select sensitive features and build feature dataset, thus reducing redundant information and improving computational efficiency.
The main innovations of this research include the following two aspects:
(1) RCMRDE is proposed for the first time, and its advantages in fault diagnosis are explored. Simulation and experimental results indicate that RCMRDE exhibits outstanding performance compared with several existing entropy.
(2) There are few studies based on VMD and JMIM feature selection. JMIM feature selection can effectively calculate the resolution of each feature and select RCMRDE with high sensitivity to construct fault feature set.
The effectiveness of proposed method is fully demonstrated by the simulation signals and actual bearing diagnosis experiment. Furthermore, the proposed method shows a superior performance compared with other machine learning methods and feature extraction methods. The comparisons show that the proposed method achieves outstanding diagnostic results, and its recognition accuracy is 97.33%.
I consider the article presented as original and contains an adequate amount of literature used.
The current scientific literature is used in the article.
In the presented article, the images have poorer graphic quality and readability of values and it would be appropriate to improve them.
Author Response
Firstly, the authors are very grateful to your comments for the manuscript. Your comments are really thoughtful and helpful. Response to the comments is given as follows. We have carefully answered each question in accordance with your comments and have carefully revised the article, all of which are marked with the “Track Changes”function.
Point 1: In the presented article, the images have poorer graphic quality and readability of values and it would be appropriate to improve them.
Response 1: Thanks for the kind remind. According to the comments of Reviewer, we have improved the graphic quality of all figures in the text, where dpi for Figures 6 and 7 is set at 300, dpi for Figures 5, 8, 9, 10 and 12 is set at 600, and dpi for Figures 1, 2, 3, 4, 11, 13, 14 and 15 is set at 1200.
We thank the reviewer again for the constructive comments and hope the above illustration can meet with the editor and reviewer’s kind approval.

Reviewer 2 Report
In this research, a novel intelligent diagnosis of rolling bearings combining RCMRDE and RF model is proposed. First, aiming at the disadvantage of traditional multi-scale entropy cannot accurately extract the useful feature from non-stationary fault signal, RCMRDE is created to characterize different fault types and damage degrees of rolling bearings. The simulated experiment shows that RCMRDE performs better in mutation signal detection, noise robustness test, stability and signal discrimination. Secondly, the JMIM method is applied to select sensitive features and build feature dataset, thus reducing redundant information and improving computational efficiency. Research results prove that a better feature extraction technique is proposed, which can effectively overcome the deficiency of existing entropy and significantly enhance the ability of fault identification. Overall, the paper structure is complete. But the following points need to be modified:
Figures 1~4 are not clear.
Author Response
Firstly, the authors are very grateful to your comments for the manuscript. Your comments are really thoughtful and helpful. Response to the comments is given as follows. We have carefully answered each question in accordance with your comments and have carefully revised the article, all of which are marked with the “Track Changes”function.
Point 1:Figures 1~4 are not clear.
Response 1: Thanks for the kind remind. According to the comments of Reviewer, we have improved the graphic quality of Figures 1-4 with their dpi set to 1200.
We thank the reviewer again for the constructive comments and hope the above illustration can meet with the editor and reviewer’s kind approval.

Reviewer 3 Report
The manuscript “Intelligent diagnosis of rolling element bearing based on refined 1 composite multiscale reverse dispersion entropy and random forest” presents a condition monitoring approach for fault detection in rolling bearings. The method involves Random Forest (RF) as a Machine Learning approach and an entropy measurement (specifically, refined composite multiscale reverse dispersion entropy, or RCMRDE) as a damage-sensitive feature.
The RCMRDE values are not extracted directly from the whole signal but rather from its band-limited Intrinsic Mode Functions (IMFs). The joint mutual information maximization (JMIM) algorithm is then used to select the 15 most sensitive values.
The RCMRDE is clearly benchmarked against two closely-related alternatives, the refine composite multiscale dispersion entropy (RCMDE) and multiscale dispersion entropy (MDE), and found superior (i.e., more sensitive to damage) than both the other candidates.
The importance of using a decomposition technique rather than extracting the damage indicators directly from the whole signal is also clearly presented and supported with data.
Therefore, the paper is of good interest for researchers in Structural Dynamics and can be definitely considered for publication. However, some major and minor issues, involving both the format and the content of the paper, should be fully addressed before acceptance.
- The paper sometimes uses the term ‘modes’ referring to the VMD-extracted Intrinsic Mode Functions. The term ‘modes’, in the context of Structural Dynamics, is too vague and might cause misunderstandings with the structure eigenmodes. Thus, the more specific terms IMF (utilised in the text as well, even if never written in full) should be used everywhere.
- Several typologies of entropy measurements have been recently used for damage detection purposes e.g. in https://doi.org/10.3390/app11135773 and other related or similar works. Apart from the already-cited sample, permutation, fuzzy, and dispersion entropies, these applications include the Rényi entropy, the Wiener entropy, and the Shannon Spectral entropy, both in their time-independent (stationary) and non-stationary forms (i.e. as Instantantaneous Spectral Entropy, or ISE, proposed for the condition monitoring of wind turbines). These closely-related features and applications should be added in the manuscript’s state of the art.
- The entropy of the output signal does not only depend on the system from which it is generated but also from the input driving force as well. In the examined case, the input was a constant; therefore, any variation in the recorded output can be directly and without doubt be linked to a structural change of the mechanism under investigation. However, how would the algorithm perform for a varying input (both in frequency and in amplitude), in absence of fault?
- While it is widely known that VMD is an efficient and reliable signal decomposition method, the motivations for preferring it over other potential data-adaptive candidates should be discussed more in detail. These include the Empirical Mode Decomposition (correctly mentioned and discussed in the state-of-the-art review) but also other algorithms such as e.g. the Hilbert Vibration Decomposition (HVD). In this regard, doi.org/10.3390/s21051825 compared VMD to EMD and HVD and other methodologies to perform SHM under time-varying conditions.
- In the introduction, it is (correctly) stated that a limitation of RSSD is the algorithm reliance on parameters that need to be set in advance. However, the VMD has a similar issue. The number of multivariate modulated oscillations K needs to be predefined by the user. This can be an issue for complex systems where the exact number of relevant IMFs is not known a priori.
Here, it is said that “The modal number is based on the decomposition criterion that the frequency center frequency of each component is not overlap […] Considering the frequency spectrum of bearing vibration signals, the model decomposition number is set as 6.”. This methodology is not clear and should be more detailed.
Furthermore, from a practical point of view, this method implicitly limits the application of the whole algorithm to structures and mechanisms with well-separated modes in the frequency domain (as for the experimental case study). This is often not the case, e.g. for symmetric buildings where the flexural modes along the main axes are generally paired and very close in the frequency spectrum of the structure response. This limitation must be clearly stated.
- In Equation 1, the intended meaning of \mu and \sigma are not explained.
- In Equation 3, the meaning of the function “Number()” is not clear.
- Equation 14, the intended meaning of the operator should be added.
- Page 9, the reference for the classic RF algorithm should be reported in the text when discussing its step-by-step implementation.
- It is not clear if the test bench QPZZ-II is retrieved from other publications or if it is an original contribution to this research; in the latter case, are the data publicly available? This is important since it would allow other authors to benchmark their proposed algorithms on the same case study, which seems to have been properly defined.
- In Section 4.4, it is not clear if the comparison is intended to highlight the efficiency of the damage index (RCMRDE after VMD), the efficiency of the Machine Learning approach (RF vs kNN; BPNN; SVM), or (most probably) both.
Minor/editorial issues:
- The authors did not use the MDPI standard format, as required. Please adjust the manuscript.
- There are several typos and grammar mistakes, e.g. “The modal number is based on the decomposition criterion that the frequency center frequency of each component [does] not overlap.” Please double-check carefully the whole text.
- The article is overall well-written but its structure is somehow peculiar. While it is clear the meaning of Section 2.3, the insertion of this (long) subsection here in the theoretical Section 2 feels a bit misplaced. Arguably, from a purely editorial point of view, it would be better first to discuss all the theoretical recalls (including VMD), present the complete procedure, and then after that present the preliminary numerical study for the comparison between MDE, RCMDE and RCMRDE; alternatively, this comparison can be kept in the theoretical discussion but should be reduced to the minimum necessary
- The graphic quality of Figures 1, 2, 3, 4, 5, 7 8, 9, 10, 11, 12, 13, and 14 is low. Please replace them with higher-quality images.
- Tables 1 and 2: please add the measurement units to all parameters.
- Table 2, since T_alpha (1/73) can be very easily derived from M_1 (73), it is not strictly needed and can be omitted.
Author Response
Firstly, the authors are very grateful to reviewer’s comments for the manuscript. Reviewer’s comments are really thoughtful and helpful. Response to the comments is given as follows. We have carefully answered each question in accordance with reviewer’s comments and have carefully revised the article, all of which are marked with the “Track Changes”function.
We thank the reviewer again for the constructive comments and hope the above illustration can meet with the reviewer’s kind approval.

Reviewer 4 Report
The paper presents an original method for rolling bearings diagnosis based on vibrational signals. The originality is obtained by combining variational mode decomposition (VMD), joint mutual information maximization (JMIM) algorithm, refined composite multiscale reverse dispersion entropy (RCMRDE), and random forest (RF) machine learning algorithm.
Also, the results are based on experiments carried out by using cylindrical roller bearings with simulated faults (inner race, outer race, and rolling elements faults).
The paper is original and I enjoyed reading it. There are a few points that need to be revised, however.
My suggestions are as follows:
- The tested roller bearings have rollers, and not balls. The word "balls" must be replaced by "rolling element" all over the paper.
- The quality of the presented Figures must be increased before publication.
- In Section 4.1, the type and the characteristics of the used acceleration sensor should be specified.
- Is there any recommendation in literature for the use of 70%-30% of the sample data for training and testing of RF algorithm? The same question remains valid for the chosen values RCMRDE parameters.
- What Fisher means and what LS means?! The authors should cite a reference presenting these methods.
- In line 363, the last paragraph above Figure 15, should mention that the observation is valid just for the KNN and RF methods.
Author Response

(The authors gave the same response as above.)

Round 2
Reviewer 3 Report
The authors of the manuscript “Intelligent diagnosis of rolling element bearing based on refined composite multiscale reverse dispersion entropy and random forest” have addressed all the main theoretical concerns raised by this Reviewer.
They also performed the minor corrections to the text, figures, and tables as suggested. The current version of the manuscript is, therefore, suitable for publication
There are only a few remaining editorial issues, which should be fixed:
- Page 9 line 264, add a blank space before the bracket: “intrinsic mode functions (IMFs),”
- Similarly for many references, e.g. “(WT)[5]”, “(RSSD)[6]”, “hilbert vibration decomposition(HVD)[7]”, “(EMD)[8]”, “network [18]”, “entropy [24]”, “entropy [25]”, “entropy [26]”, “entropy [27]”, “In [36]”, “steps[47]”, “LS[49]”, and elsewhere throughout the whole text,
- In ““hilbert vibration decomposition”, Hilbert should be capitalised as it is a surname.
- The figures depicted in the Reviewer’s pdf version are still blurred. However, this might be due to the automatic conversion from .docx to .pdf. This can be fixed by uploading them separately on the MDPI platform, such that the published article will have good quality figures.
- Page 6 line198, the font style from “m=2” to “n=2048” seems to be different from the rest of the text and the other in-line equations.
Author Response
Firstly, the authors are very grateful to your comments for the manuscript. Your comments are really thoughtful and helpful. Response to the comments is given as follows. We have carefully answered each question in accordance with your comments and have carefully revised the article, all of which are highlighted.
